# Melatonin Interaction with Other Phytohormones in the Regulation of Abiotic Stresses in Horticultural Plants

**DOI:** 10.3390/antiox13060663

**Published:** 2024-05-28

**Authors:** Shanxia Huang, Songheng Jin

**Affiliations:** Jiyang College, Zhejiang A&F University, Zhuji 311800, China; huangshanxia@zafu.edu.cn

**Keywords:** melatonin, growth traits, antioxidants, stress mitigants, photosynthesis, redox balanced

## Abstract

Horticultural crops play a vital role in global food production, nutrition, and the economy. Horticultural crops are highly vulnerable to abiotic stresses. These abiotic stresses hinder plant growth and development by affecting seed germination, impairing photosynthetic activity, and damaging root development, thus leading to a decrease in fruit yield, quality, and productivity. Scientists have conducted extensive research to investigate the mechanisms of resilience and the ability to cope with environmental stresses. In contrast, the use of phytohormones to alleviate the detrimental impacts of abiotic stresses on horticulture plants has been generally recognized as an effective method. Among phytohormones, melatonin (MT) is a novel plant hormone that regulates various plants’ physiological functions such as seedling development, root system architecture, photosynthetic efficiency, balanced redox homeostasis, secondary metabolites production, accumulation of mineral nutrient uptake, and activated antioxidant defense system. Importantly, MT application significantly restricted heavy metals (HMs) uptake and increased mineral nutrient accumulation by modifying the root architecture system. In addition, MT is a naturally occurring, multifunctional, nontoxic biomolecule having antioxidant properties. Furthermore, this review described the hormonal interaction between MT and other signaling molecules in order to enhance abiotic stress tolerance in horticulture crops. This review focuses on current research advancements and prospective approaches for enhancing crop tolerance to abiotic stress.

## 1. Introduction

Abiotic stressors pose significant risks to plant development and production in agriculture, negatively impacting crops. Abiotic stressors impede vegetative growth and diminish the production of crops [1]. Abiotic stressors have the potential to decrease the productivity of plants by 50 to 70% [2]. As a result of global warming and changing climatic situations, abiotic variables have become worse and more unpredictable [3]. Abiotic stresses impaired various plant physiological functions, causing effects such as hindered seed germination, damaged lateral root development, altered secondary metabolites production, increased heavy metals (HMs) accumulation, decreased mineral uptake and relative water content, and changes in antioxidant defense mechanisms in plants, all of these parameters collectively leading to plant death [4,5,6]. Plants face a wide range of abiotic stresses throughout their entire life cycle. Because they are sessile, plants have developed a wide range of response mechanisms to deal with challenging environments and ensure their survival and ability to reproduce [7]. Several approaches have been suggested to enhance productivity in plants under stress. Recent research has shown that exogenous phytohormone treatment is a useful strategy for reducing the adverse effects of stress in plants [8,9,10]. Phytohormone-like plant growth regulators (PGRs) such as melatonin (MT), salicylic acid (SA), brassinosteroids (BRs), gamma-aminobutyric acid (GABA), jasmonic acid (JA), gibberellin (GA), cytokinins (CKs), auxin (AUX), and other growth regulators have shown significant potential to enhance abiotic stress tolerance in horticultural plants [11,12,13]. Phytohormones regulate the seedling health index, seed germination rate, photosynthetic apparatus, chlorophyll content, proline, soluble sugar content, metabolites uptake, redox homeostasis, flowering, leaf senescence, root development, later and adventitious root development, and antioxidant defense system, and they increase mineral uptake and enhance abiotic stress tolerance in horticultural plants (Figure 1) [14,15,16,17]. Phytohormone-like growth regulators enhanced salinity stress tolerance in tomato [18], water stress in pepper [19], HMs stress in pepper [20], cold stress in spinach [21], and heat stress in cucumber [22]. 

MT is a dynamic molecule having low molecular weight [23]. In addition, MT is a novel PGR that has strong antioxidant properties [24]. Furthermore, MT played a significant role in the regulation of photosynthetic machinery [25]; modification of root system architecture [26], antioxidant pool [27], mineral nutrient accumulation [28], flowering [29], fruit ripening [13], circadian rhythms [22]; and improvement of plant growth and development [30]. In addition, MT positively regulated the growth status of many plant species under salinity [31], drought [32], heat [33], cold [34], alkaline [35], and HM stress [20]. MT can efficiently eliminate various reactive oxygen species (ROS) through boosting the activity of antioxidant enzymes and the upregulations of antioxidant genes level, and it can protect plants from oxidative damage [36]. In addition to its antioxidant properties, MT significantly altered the expression of several genes involved in physiological functions [37]. 

This review explores the significant effect of MT on several aspects of plant physiology, explicitly emphasizing its impact on growth, antioxidant enzyme activities, mineral uptake, photosynthetic capacity, and regulation of gene expression. In addition, this review describes recent advancements and the potential role of MT in enhancing stress tolerance and explores effective strategies for using exogenous MT to mitigate damage caused by abiotic stress. Further, MT’s interaction with other phytohormones is also described. 

Briefly, we analyzed and discussed the role of MT in regulating plant growth. We aim to answer the following research questions: (i) What are the physiological mechanisms through which MT regulates plant growth and development? (ii) How does MT interact with SA, BRs, GABA, JA, GA, CKs, and AUX to regulate plant growth? (iii) Are there any overlooked physiological traits crucial for understanding MT’s role in stress regulation? We discussed the recent advancements in this area of research, distinguishing this review from previous ones. This study provides researchers and policymakers with valuable insights to develop effective strategies for mitigating the impact of abiotic stress on horticultural crops, thereby contributing to global food security enhancement.

## 2. Yield Losses in Horticultural Crops: A Threat to Food Security and Nutrition

MT contributes to the level of endogenous hormones that plants produce to regulate their growth in response to oxidative stress biomarkers. These play a crucial role in the integration of several pathways that transmit signals under abiotic stressors. These also contribute to the regulation of many responses necessary for the characterization of tolerant inbred lines. The importance of MT as a signaling molecule in stress tolerance has been previously investigated by many plant researchers [38,39]. These phytochemicals also play a crucial role in sustainable agricultural production. The regulatory function of phytohormones in many plants’ developmental processes, including physiological and molecular activities, has been emphasized in published findings on the adaptation to oxidative harms [40]. Plants encounter a wide range of disturbances, starting from seed germination to throughout their whole life cycles. Numerous abiotic stressors and challenges result in crop losses by lowering the quality and yield of crops. Abiotic stresses alter morpho-physiological processes, which have an influence not only on productivity but also the quality of the produce. Numerous abiotic stressors are also a result of recent climatic changes for fruit crops [41]. A significant problem that the agricultural industry is facing is the climate crisis. Plants have been found to exhibit several stress reactions, which include a loss in the productivity of the photosynthetic system, relative water content, membrane damage, photosynthetic process, plant development, and yield. Furthermore, 90% of agricultural fields are affected by one or more stress factors [42]. Thus, the horticulture industry is actively looking for innovative agronomic techniques that can counter the negative effects of environmental conditions while ensuring productivity and sustainability. In this way, horticultural crops are safeguarded from abiotic stimuli by a wide range of plant hormones. Endogenous phytohormones are described as important mediators of plant reactions under stress circumstances, in addition to their functions in developmental changes and tolerance to abiotic stress [43]. MT is a small signaling molecule that affects practically every element of plant growth and yield (Table 1). Different physiological mechanisms can have divergent action mechanisms with phytomelatonin. As a result, it has been found that a single hormone can occasionally have cell-wide control and can devise strategies, whereas numerous hormones may act simultaneously to control the same activity. Plants need several phytohormones exogenously to support and manage themselves against oxidative harms by improving the plant defense system. Among these phytohormones, MT is multifaceted and contributes to the tolerance of plants subjected to environmental stressors. Phytohormones improve abiotic stress tolerance in horticulture crops. Hence, MT spraying is performed to improve future crop stress management studies [44].

## 3. Multifunctional Role of MT in Horticultural Plants

MT is a multifunctional molecule in plants under stress environments [19]. MT potentially increased stress tolerance by upregulating the antioxidant defense mechanism, protecting the photosynthetic system; increasing mineral nutrient fluxes; improving proline, soluble sugar, and secondary metabolites production; balancing redox homeostasis; maintaining mineral homeostasis; increasing relative water content; and promoting root architecture system and increased growth and development in response to stressful environments [13,54]. In addition, MT has a broad spectrum and has naturally occurring antioxidant molecules. Furthermore, the potential functions of MT in plants are described in Table 2.

### 3.1. Stress Mitigation in Horticultural Crops: Impacts of MT

Several studies have suggested that MT is a well-known plant hormone in response to stress conditions (Figure 2) [28]. MT regulated seed germination in cucumber under salinity stress [62], upregulating nitrogen metabolism in pepper under drought stress [63], modulated the antioxidant defense mechanism in *Camellia sinensis* L. under cold stress [64], reduced oxidative damage and enhanced the defense mechanism in strawberry under heat stress [65], increased photosynthetic efficiency in sweet potato under chromium toxicity [66], and increased mineral nutrient accumulation in tomato under nickel toxicity [67]. Consequently, the investigation of exogenous MT in stressed plants, with regard to plant adaptability and survival, has been receiving significant attention from researchers.

### 3.2. MT and Salinity Stress

Salinity stress is a particularly important abiotic stress that negatively affects crop productivity on a minimum of 20% of irrigated land globally [68]. Salinity not only triggers water deficiency due to osmotic stress but also disrupts crucial metabolic processes such as the leaf photosynthetic mechanism, alters the antioxidant system, and causes oxidative damage [54]. Recently, Altaf et al. [13] reported that MT played a significant role in response to stress conditions. In another study, salinity toxicity was considered to decrease the growth traits, photosynthetic content, and potassium uptake in tomato plants in contract; MT application significantly increased the growth traits, protected the photosynthetic system, increased pigments molecules, improved potassium accumulation, and reduced sodium ion uptake in tomato under salinity conditions [69]. MT supplementation considerably improved biomass production, protein content, and antioxidant enzyme activity in cucumber plants under salinity stress. In addition, MT application effectively decreased ROS molecule production, reduced MDA uptake, lowered EL level, and reduced oxidative damage in cucumber leaves under a salinity stress environment [70]. In watermelon seedling leaves, salt stress reduced photosynthesis-accelerated ROS production and damaged cell membranes. In contrast, MT pretreatment of roots dose-dependently reduced salinity-induced reductions in photosynthetic rate and oxidative damage [71]. Furthermore, Efimova et al. [72] observed that MT treatment showed positive effects on the uptake of secondary metabolites accumulation, proline uptake, and antioxidant enzyme activity, leading to a reduction in oxidative damage in potato leaves. Previous studies have reported that MT significantly enhanced salinity stress tolerance in strawberry [73], apple [74], and fenugreek [75]. MT can mitigate the adverse effects of salt stress on horticultural crops.

### 3.3. MT and Drought Stress

Climate change has caused drought stress, which reduces agricultural productivity both in terms of quality and quantity [76]. Water is a vital component for a proper plant. Drought often refers to a situation when the available amount of irrigation water is insufficient to meet the demand [77]. Drought stress alters plant metabolic functions. In contrast, MT positively protects plant metabolic processes such as maintaining photosynthetic activity, repairing root architecture systems, and regulating antioxidant enzyme activity [78]. The application of MT significantly reduced the MDA uptake and decreased chlorophyll degradation and ROS uptake in fenugreek leaves. Conversely, MT application potentially increased the growth parameters, chlorophyll content, and endogenous MT concentration in fenugreek leaves under drought stress [79]. Foliar application of MT remarkably improved the yield and quality-related traits of M. oleifera under drought environment by increasing secondary metabolites production, pigments content, and the antioxidant defense mechanism while reducing MDA, EL, and ROS accumulation [80]. In another study, Mushtaq et al. [81] reported that MT supplementation markedly increased root morphological traits, photosynthetic-related parameters, soluble sugar, and proline content in tomatoes under a drought stress environment; in addition, MT effectively reduced excessive ROS uptake and MDA and El levels and reduced oxidative damage in tomato leaves. MT efficiently enhanced the drought stress tolerance in pepper by increasing the endogenous MT level in pepper leaves [82]. MT foliar spray is an effective method for improving the growth and biomass production of cauliflower grown under drought stress [83]. MT alleviates the negative effects of drought-induced oxidative damage in potato plants via the regulation of photosynthetic pigment content, secondary metabolite accumulation, and antioxidant defense mechanism and via the increase in tuber yield of potato plants under a drought stress environment [84]. The growth of strawberry seedlings was severely impeded by drought stress. In contrast, MT application considerably improved photosynthetic assimilation rate, chlorophyll content, and proline uptake and reduced oxidative damage in strawberry seedlings under drought stress environments [85].

### 3.4. MT and Cold Stress

Cold stress has a significant negative impact on plant growth and productivity worldwide [86]. Low temperatures can harm and even cause the death of crop species, which may have a negative impact on their ability to produce crops [87]. MT prominently enhanced cold stress tolerance in tomato [88], pepper [89], watermelon [90], melon [52], and cucumber [34]. MT has a crucial role in enhancing the antioxidant capacity and maintaining the redox balance in tomato plants when they are exposed to cold stress. Additionally, MT promotes photosynthesis in these plants under chilling stress [91]. The application of MT alleviated the decrease in photosynthetic ability caused by cold exposure by lowering oxidative damage via increased antioxidant potential and maintaining redox homeostasis. The results revealed that MT has the potential to mitigate the negative effects caused by cold temperatures in tea plants [64]. Altaf et al. [25] observed that MT application efficiently protected the photosynthetic performance (increased chlorophyll content, pigments content, and related gene expression analysis) of pepper seedlings under cold stress. Exogenous MT has the potential to enhance the development of potato seedlings, decrease the levels of MDA and ROS accumulation in leaves, stimulate the production of secondary metabolites and proline, boost the antioxidant enzymes activities, and promote the non-enzymatic antioxidant enzymes activity in potato leaves [92]. The findings demonstrated that MT supplementation effectively mitigated the growth inhibition induced by cold stress. This was evidenced by the enhanced plant growth and reduced excessive ROS molecule production and MDA concentration. Surprisingly, the application of MT increased the enzymatic and non-enzymatic antioxidant enzyme activity in melon seedlings cultivated under cold stress. In addition, MT application significantly enhanced the levels of proline and soluble protein in response to cold stress. The findings indicate that MT has a beneficial impact on melon seedlings by protecting them from cold stress [93]. Pretreated strawberry plants with MT improve protection against cold stress by increasing their antioxidant defense capability and regulating the DREB/CBF–COR pathway [94].

### 3.5. MT and Heat Stress

Heat stress is characterized by a prolonged increase in temperature that exceeds a certain threshold level, resulting in long-term negative effects on plant growth and productivity [95]. Heat stress adversely affects seed germination, leaf photosynthesis, and root development [96]. Heat stress is considered to decrease the growth of many plant species. In contrast, MT effectively regulates the heat stress tolerance in many plant species [97,98,99]. MT pretreatment improved net photosynthetic rate, pigments concentration, and photosynthetic efficiency; protected chlorophyll florescence’s characteristics; and decreased photo inhibition in tomato leaves under heat stress [100]. Furthermore, MT application reduced oxidative damage by reducing oxidative stress biomarkers such as H_2_O_2_, superoxide ion (O_2_^•−^), MDA, and EL in tomato leaves under heat stress. In addition, MT effectively increased the antioxidant enzyme activity and photosynthetic apparatus in tomato seedlings under heat stress [101]. MT application on celery plants substantially improved the plant’s capacity to eliminate ROS molecules generated in response to heat stress, thus enhancing the plant’s resistance to heat stress [102]. The leaf photosynthetic performance, antioxidant enzyme activity, and chlorophyll content were increased, and oxidative stress biomarkers were decreased in potato leaves by the application of MT under heat stress [103]. Xing et al. [104] noticed that MT treatment regulated chlorophyll metabolism, secondary metabolites metabolism, sucrose and starch metabolism, and carotenoid biosynthesis-related gene expression in chrysanthemums. Importantly, MT potentially enhanced heat stress tolerance in chrysanthemum plants. The endogenous MT is crucial for its ability to act as an antioxidant, and it is also crucial for maintaining redox balance and upregulated antioxidant defense mechanism in tomatoes under a heat stress environment [105]. MT application significantly increased fruit yield-related traits, photosynthetic capacity, and chlorophyll content in cucumber leaves under heat stress. In addition, MT effectively reduced oxidative damage by reducing the MDA concentration and EL level in cucumber leaves [36].

### 3.6. MT and HMs Stress

In recent times, HM toxicity has emerged as a worldwide concern, presenting a significant risk to the environment [106]. Plants have negative effects on their cellular metabolism when they are exposed to a greater concentration of HMs [107]. HMs toxicity impaired plant metabolic functions, causing effects such as decreased photosynthetic efficiency, restricted mineral accumulation, increased HMs uptake, and eventually a decline in plant growth and sudden death [108,109]. On the other hand, MT supplementation potentially improved HMs stress tolerance in pepper [20], sweet potato [66], cucumber [110], and tomato [67]. In a recent study, Saqib et al. [111] reported that MT treatment effectively increased the growth attributes, photosynthesis, and secondary metabolites content and decreased the MDA content, EL level, and ROS production in two pepper species under nickel toxicity. Furthermore, MT treatment restricted chromium accumulation and increased mineral nutrient uptake by increasing the root system architecture in pepper under chromium toxicity [112]. In another study, Nawaz et al. [113] observed that MT-pretreated watermelon seedlings showed significant improvements in root morphology, antioxidant enzyme activity, and net photosynthetic rate and reduced the H_2_O_2_ level, MDA content, and vanadium uptake in watermelon leaves. MT application significantly enhanced growth and antioxidant enzyme activity in cucumber under cadmium toxicity [114], and MT application affected nickel accumulation in tomato [67], cadmium uptake in strawberry [59], boron accumulation in spinach [115], and vanadium accumulation in muskmelon [116]. MT supplementation dramatically increased the photosynthetic assimilation rate, chlorophyll content, and antioxidant enzyme activity and reduced excessive iron accumulation in cucumber leaves under iron toxicity [117]. In another study, Saqib et al. [118] found that MT application combined with cadmium increased growth, pigment content, and antioxidant enzyme activity in strawberry seedlings. In addition, MT treatment hindered cadmium accumulation from root to shoot. In a nutshell, several studies suggested that MT is a significant molecule that regulates HM stress tolerance in horticultural plants [11,119,120]. 

## 4. Multifunctional Role of Phytohormone in Horticultural Plants

Phytohormones are famous as plant-protecting hormones under abiotic stresses [Table 3] [50]. These are well-known management strategies that function as bioactive substances in plants for the reduction of numerous abiotic stresses [121]. Many phytohormones can be sprayed exogenously to boost plant defense mechanisms and lessen the effects of abiotic and biotic stresses, which is important for sustainable production [Figure 3] [122]. MT is more efficient at reducing the adverse effects of oxidative injury that occurs from abiotic stresses at any stage of growth and yield [123]. MT applications can support a wide range of signaling and transduction networks through regulatory functions and crosstalk with other phytohormones. Numerous important functions are perceivable in the synchronization of phytohormones that deal with the detrimental effects of ROS, ionic channels, and mitogen-activated protein kinase (MAPK) signals that occur from abiotic stressors [124].

To develop tolerant germplasm, it may be useful to comprehend the interaction between phytohormones and microarrays, as revealed by Sharma et al. [125]. Plant yield was observed to increase with the modulation of physiological and biochemical responses cultivation under abiotic stresses. Numerous sensitive landraces may also have higher yields with sufficient supplementation of phytohormones based on the genetic makeup and environmental conditions of the characterized germplasm [126]. However, it is necessary to understand the beneficial role of phytohormones in the regulation of endogenous hormonal regulations in plants [127]. However, genetic improvement techniques are also efficient for the development of tolerant landraces [128]. MT supplementation is a more efficient and rapid way to improve abiotic stress tolerance in horticultural crops, as reported by earlier plant researchers [122].

**Table 3 antioxidants-13-00663-t003:** The role of exogenous phytohormones against abiotic stresses in vegetable production.

Phytohormones	Stress	Vegetables	Key Findings	References
JA	Drought	Potato	Reduced relative leaf water potential was the result of *StJAZ1* overexpression in plants. MDA was improved. JA is useful for lowering MDA production. and lipid peroxidation.	[129]
SA	Heat stress	Potato	An amount of 2, 3, 8, 14, and 21 dpi was applied on Chicago and Gala cultivars, resulting in improved physiological and biochemical processes	[130]
ABA	Salinity	Lettuce	A concentration of 0, 100, 200, 300, and 400 mg L^−1^ increased lettuce output and performance.	[46]
MT	Drought	Cucumber	A concentration of 100 μM suggestively enriched growth, yield, and defense mechanism.	[51]
SA	Drought	Tomato	A concentration of 10–5 M improved growth under 10 days and water holding capacity.	[131]
SA	Chilling injury	Sweet potato	By increasing plant antioxidant capacity, SA treatment reduced the likelihood of chilling harm.	[132]
Osmoprotectants	Drought	Tomato	A total of 50–57% of the field capacity also increased the level of salts in plants. Germination was very poor in seeds.	[46]
SA	UV-radiation	Pea	A concentration of 0.4 mM enhances growth by improving the defense system.	[133]
Polyamines (PAs)	Salinity	Cucumber	The application of PAs such as spermidine can be controlled when harmful effects from 50 mM NaCl arise.	[47]
SA	HMs	Melon	The amount of Cd adsorption, excessive ROS production, the level of proline, the amount of protein, and the level of oxidative enzymes all decreased at the required 0.1 mM concentration.	[134]
SA	Chilling injury	Sponge gourd	A concentration of 1.5 mM L^−1^ was found to be more effective for the reduction of chilling injury under storage conditions.	[135]
SA	Heat stress	Cucumber	The plant defense system was strengthened by exogenous spray (1 mM) by lowering MDA, H_2_O_2_, and ROS. SA thereby boosted the amount of photosynthesis that was disturbed due to temperature extremities.	[128]
BRs	Salinity	Peppermint	Antioxidant enzyme system maintained cellular membrane integrity, and secondary metabolites generation.	[136]
Epibrassinolide	Cold stress	Cucumber	Better seedling health index, increased chlorophyll content, antioxidant enzyme activity, upregulated gene expression, and reduced oxidative stress level.	[137]
BRs	Salinity	Lettuce	Lessened the adverse effect of salinity by reduced oxidative injury and improved antioxidant enzyme action.	[138]
BRs	Cadmium	Radish	Improved seed germination level, proline meditation, antioxidant enzymes, and fresh seedling weight of radish.	[139]
Brassinolide	Lead	Radish	Reduced oxidative stress in plants growing under leaf toxicity. Moreover, antioxidant enzymes were also activated.	[140]
SA	Heat	Strawberry	Improved plant growth under stressful conditions. Better crop performance under heat stress. Activation of plant defense system.	[141]
SA	Boron toxicity	Watermelon	Better melon plant performance under boron toxicity. Regulation in the photosynthetic system of plants. Lessening of oxidative harms.	[142]
SA	Chromium	Tomato	Regulation of photosynthetic pigment under chromium toxicity. Improved plant growth by regulation of oxidative damage.	[143]
MT	Drought	Pea	Reduced oxidative stress in plants growing under water stress. Antioxidant enzymes were also activated.	[144]

### 4.1. MT Interacts with Other Phytohormones in Horticultural Plants

The supplementation of MT and its interplay with other phytohormones is a more effective strategy for the reduction of adverse effects of abiotic stresses to robust yield [15]. Plant researchers are paying huge attention to the role of phytohormones due to their versatility in the face of abiotic stresses [27]. Their application is still rather restricted in stressed plants. Therefore, the current review examines in more detail the phytohormones that can be sprayed on horticulture crops subjected to abiotic stressors [13]. Moreover, deep insights into biochemical, physiological, and molecular bases were also explored to combat the negative effects of abiotic stressors that disturbed the productivity of horticultural crops. 

Several plant researchers revealed interactions between MT and other phytohormones and their significant impact on the plant sustainable productivity of horticultural crops [20]. Due to their chemical similarities, MT is a master PGR and stimulator that has antioxidant properties [145]. Furthermore, MT has the ability to upregulate genes that are responsible for producing GAs that stimulate the development of new shoots [71]. MT may enhance the expression of genes related to the production and transmission of CKs, which are known to stimulate the development of shoots in plants [146]. MT also regulates the ABA signaling system by controlling the expression of genes involved in ABA production and signaling [147]. Li et al. [148] reported that MT has been shown to increase AUX biosynthesis gene expression and enhance AUX transport in apple seedlings, hence promoting adventitious root development. Furthermore, increased expression of BR biosynthetic genes, along with greater concentrations of BRs, was observed in Chinese hickory plants treated with MT supplementation under drought stress [149]. In addition, exogenous MT enhances the expression of BR biosynthetic genes under abiotic stress, resulting in an increase in endogenous BR levels and primarily increasing the stress tolerance [150,151]. The elevated concentration of JA, induced by exogenous MT, played a vital role in promoting seed germination and maintaining development in plants [152]. However, this process was considerably hindered when the seedlings were exposed to saline stress. Several investigations have found intriguing links with MT and almost all recognized plant hormones, such JA, SA, BRs, PAs, and strigolactones, as well as growth regulators like AUX, GA, CKs, and ABA [38,42]. 

### 4.2. MT and SA 

This hormone is the phenolic-based compound that helps plants defend themselves for appropriate growth and yield under climate-related extremities [128]. Different SA levels were exogenously applied to a pea crop under induced salinity (150 mM NaCl). SA was found to enhance pea growth, yield, defense activities, and compatible solutes in this research activity [133]. Ahmed et al. [153] reported that SA effectively increased the seed germination rate, chlorophyll content, antioxidant enzyme activity, and nutrient use efficiency in sweet peppers. Overall, SA has been proven to be more beneficial for horticultural crops cultivated under abiotic stresses. Similar to this study, 0.11 mM SA also increased potato’s resistance to cold stress [154]. Therefore, SA is a good strategy for horticultural crops to tolerate abiotic stressors. The supplementation of 1 mM SA increased the diffusion of gases, good water usage potential, and production of enzyme reactions and decreased cell damage under heat stress [129]. Furthermore, biomass on a fresh or dry basis was reduced, which ultimately resulted in a loss in production due to stressful conditions. Osmotic stress also disrupts photopigments, photosynthesis, and stomata function [41]. The uniformity in the photosynthesis process and stomatal functionality are significant with a supplemental spray of SA [33]. It has been revealed that SA can reduce oxidative injury and osmotic damage, which helps to lessen the negative effects of abiotic stress [155].

SA regulates biological stress reactions in plant tissues [156,157]. MT supplementation boosts the manufacturing of SA, JA, and ethylene. Previous research [28] established the importance of MT in plant defense against biotic stressors. A combined application of MT and SA effectively promoted tomato growth, maintained photosystem functions, balanced the ascorbate and glutathione enzymes cycle, decreased methylglyoxal enzyme activity, and upregulated glyoxalase enzyme activity in tomato under drought stress environments [32]. Exogenously provided MT defended plants from the tobacco mosaic virus by boosting the synthesis of SA and nitric oxide. MT exposure reduced virus titer levels [48]. Hence, it has been studied that MT and ethylene are effective for improving endogenous SA in plants to cope with the negative effects of abiotic stressors in horticultural crops (Table 4).

### 4.3. MT and JAs 

Jasmonates are plant hormones that include methyl jasmonate (MeJA) and JA. They regulate a wide spectrum of crop productivity and stress responses [12]. MeJA and JA have recently been investigated for their effects on horticulture crops. The restriction of chlorophyll activity and tuber development was controlled due to the uptake and translocation of JAs. Exogenous application of PGR enhanced sugar beet development and drought resistance [129]. Moreover, the introduction of exogenous JAs exogenous spray increased the endogenous level of JAs production. MeJA increased bioactive molecules, which facilitated cauliflower plants to be more drought-resistant. Therefore, it has been demonstrated that adding JAs can boost crop yield [155]. JAs improve the defense mechanism against environmental stressors. These aspects are important for horticultural crops that are grown in abiotic stressed environments [160]. As a coping strategy for vegetable crops over abiotic stress, JAs have great potential for sustainable productivity. Although ROS are a sign of stress in tomato plants, JAs can reduce the generation of ROS. MeJA demonstrated a good ability to improve the productivity of tomato landraces that were subjected to high saline levels [161]. The indigenous hormonal concentrations of JAs were improved when MeJA was sprayed exogenously on peas growing in challenging conditions. Cauliflower faces difficulties regarding development at the vegetative stages and poor yield at the reproductive phase when there is a water shortage. MeJAs may have stimulated enzymatic and non-enzymatic processes. The external use of MeJAs as a supplement reduced the assimilation and absorption of toxic metals in eggplant [162]. Consequently, it has been noted that JA helped to improve osmoprotectant levels, defensive activities, photopigments, and ROS elimination, decrease H_2_O_2_, and lower MDA levels towards salt. JAs also contributed to increasing the sprouting of okra seeds and the number of seedlings [163]. MeJAs applied exogenously to peppers enhanced the production of osmolytes, bioactive compounds molecules, metabolism, and mineral absorption through roots. Additionally, by adding MeJA to the mix, reductions in MDA, H_2_O_2_, ion toxicity, and free superoxides were also detected. Thus, it has been determined that JAs are an appropriate phytohormone for reducing the negative effects of abiotic stress in plants [164]. MT crosstalk with other signaling molecules is effective for plant growth and productivity (Figure 4).

JAs increased the expression of genes, proteins, and bioactive molecules involved in the plant defense system [165]. According to a recent study, the MT–JA relationship is highly complex. MT medications have an indirect effect on JA concentrations in abiotic stress studies. MT inhibits JA production and levels in *B. napus* responding under saline conditions. Moreover, it induces the generation of JA proteins (repressor proteins in the JA signaling pathway), which decrease the reaction mediated by JA. Therefore, it plays a significant role in the improvements of tolerance in plants [166]. However, an increase in JA concentrations has been observed in tomatoes grown under water stress [158]. Chromium toxicity poorly reduced the growth of tomato seedlings. In contrast, MT application along with JA positively regulated physiological traits such as chlorophyll content and nitrogen metabolism; enhanced secondary metabolites production; upregulated antioxidant enzymes activity; increased γ-glutamyl kinase activity; and reduced oxidative damage and chromium accumulation in tomato seedlings under chromium toxicity [167].

### 4.4. MT and BRs 

These are newly updated, environmentally friendly, and multipurpose plant hormones that control metabolic processes in plants. Plant biologists are encouraged to use these plant hormones to produce crops sustainably [14]. For plants thriving in abiotic stress scenarios, BRs seem to be more beneficial. According to research, BRs improved the germination rate, root penetration, seedling growth, cell elongation and specialization, fruit set, reduced leaf, and multiplication of floral components in crops [168]. Additionally, earlier research work has shown that BRs can enhance the development features, nutritive value, antioxidant potential, and osmolytes and protect against membrane damage. BRs improved pea performance in the face of abiotic stress. Similarly, it has been demonstrated that BRs successfully increase salt tolerance in horticultural crops [169]. Abiotic stress resilience can be improved by BRs in horticultural crops. Foliar applications of these phytohormones that resulted in better metabolites and increased enzymatic activity were also observed. The productivity of tomato, pepper, and cucumber crops was improved by using BRs, with enhanced physiological processes and a decrease in oxygen radicals being observed. Abiotic stress tolerance in radish with the use of supplemented BRs was improved [170]. The reduction in ROS, MDA, H_2_O_2_, and decreased ion toxicity indicates that BRs are phytohormones that relieve stress for radish plants that grow under unfavorable conditions. Furthermore, the improvement in plant defense mechanisms demonstrated that BRs are efficient in strengthening the plant immune response against challenging environments. BRs are considerably more successful at coping with abiotic stress because cucumbers showed a comparable tolerance process [171]. The spraying of 24-EBRs enhanced the exchange of gases activities and all of the linked characteristics, including starch, soluble sugars, and rubisco capabilities, in cucumber. It is considerably more effective for crops with better yields that are growing in both normal and stressful conditions. BRs potentially scavenged the free radicals that strengthened the antioxidant potential of radish against copper toxicity [172]. According to earlier research, BRs are naturally occurring compounds that might be widely used to lessen the negative effects of environmental stresses [173,174].

BRs are steroid hormones that govern plant growth and development by affecting cell dilatation and multiplication. The significance of BRs in protecting plants from osmotic adjustment is well recognized [175]. The use of MT regulates the development of BRs in plants by increasing the transcript concentration of BR-generating genes. Photomorphogenesis is regulated by phytohormones because plants respond to light signals to improve growth [30]. MT accelerated the growth of tomato seedlings by upregulating the photosynthetic capacity and reduced the oxidative damage in tomato seedlings [167].

### 4.5. MT and GABA

GABA is a signaling molecule with a well-recognized global status and multiple activities, and it is one of the non-proteinogenic amino acids in plants [17]. GABA is recognized to govern some physiological processes in horticulture crops, including the control of seedling growth, osmolytes deposition, optimum photosynthetic capability, root morphology, modulation of plant yield, and ionic, redox, and antioxidative defense systems [176]. The use of GABA during stress is reported to dramatically increase the morphological and physiological capabilities of plants, particularly the generation of proline, total sugars, PAs, biosynthesis, and chlorophyll content. This molecule is also well known for reducing the overproduction of ROS when stressed, mostly by activating the antioxidant defense system [177]. 

Exogenous GABA supplementation also markedly increased root development, appropriate ion regulation, osmolyte accumulation, stress-related protein, and leaf photosynthetic features in black pepper under PEG-induced stress [156]. In tomato seedlings under cold exposure, a foliar spray of GABA significantly increased proline concentration, relative water content, stomatal conductance, and antioxidant activity. Ion homeostasis depends on GABA permeability constituents. In addition, GABA significantly improved nutrient uptake, chlorophyll fluorescence, and antioxidant enzymes and reduced ROS production while maintaining the integrity of cell membranes in peaches under cold stress [156]. Under osmotic damage, GABA is the main mediator in inducing leaf senescence. Through the activation of the antioxidant defense system and an improvement in photochemical effectiveness during short/low light stress, GABA boosted the stress tolerance of chilies. The effects of GABA supplementation on chilies included an upsurge in aerobic respiration, net photosynthetic, and fluorescence qualities; an increase in SOD and CAT activities; and a decline in MDA activity [178]. The biosynthesis of GABA is increased to such a degree that it exceeds the cellular concentrations of non-proteinogenic amino acid compared with the amino acids involved in protein biosynthesis after environmental stressors, i.e., salt, water stress, toxic metals, and low and elevated temperatures [179]. Interestingly, higher GABA production was seen in response to stress, and a relationship between its degradation and other elements was discovered, such as oxidative stress defense, the activation of antioxidant enzymes, osmolytes management, and balancing ion homeostasis. Plants’ ability to withstand abiotic stress has been increased because of GABA activation [179]. Under calcium nitrate stress, the treatment of GABA dramatically accelerated muskmelon production. Additionally, GABA treatment greatly boosted the activity of defense-related activities. By successfully raising spermidine and spermine levels while lowering putrescine levels in the leaves, external GABA treatment was able to improve PAs production. In addition, overexpression boosts endogenous GABA concentrations in carrots, tomatoes, and peaches, which in turn enhances the enzymatic activity and pigment concentration and eventually increases plant tolerance to environmental stresses [180]. 

### 4.6. MT and PAs

The smaller molecular weight compounds spermidine, putrescine, and spermine comprised the majority of the phytohormones, such as PAs, group [13,35]. Through all the improvement of the respective vegetable’s root, leaf distinctiveness, pollen supportability, pollen tube growth, fruit growth, transcriptional regulation, biogenesis, morphogenesis, leaf senescence, somatic embryogenesis, and fruit ripening, PAs were used to manage numerous biochemical and physiological mechanisms [181]. Different PA levels can be applied to fruit and vegetable crops to control a wide range of abiotic stressors. These can be controlled by changing a variety of plant functions using a spray of PAs that are sold on international marketplaces. Furthermore, tomato seedlings cultivated under stressed environments responded favorably to the exogenous use of spermidine [182]. The number of PA molecules within cells and partitions was also increased by the use of spermidine, particularly in the plant roots at the seedlings stage of tomato. Horticultural crops that grow in saline environments may benefit from a higher content of spermidine. PA supplements can enhance the separation of ions and their transport to other plant sections. Spermidine has been discovered to help enhance plant growth, net photosynthesis, proline concentration, and other sugars [183]. It has been stated that PAs sprayed on tomato plants decreased the levels of ROS and H_2_O_2_ generation. PAs, i.e., spermine, putrescine, and spermidine, were used to enhance pepper seed germination. It has been found that treated seeds germinate more quickly, have higher germination indices, and germinate earlier than untreated seeds [184]. It was discovered in another study that adding PAs to germinating seeds enhanced crop performance in adverse conditions. As in the case for increased production in tomato, PAs are chemicals that relieve stress. Indicators of stress relief in plants include a decrease and balance in the production of ROS, H_2_O_2_, free radicals, and membrane leakage. Consequently, earlier research has shown that PAs are a multifunctional hormone-like growth regulator for enhancing endogenous hormones and enhancing the stimulation of toxic scavengers [185].

MT and PAs have now been identified in an array of physiological functions and abiotic stresses that plants encounter. The MT-induced modulation of PAs biosynthesis has been demonstrated to play a vital role in the expression of stress tolerance systems in an array of crops [186]. Exogenous MT exposure has been proven to boost endogenous PA concentrations via the upregulation of ornithine decarboxylase and arginine decarboxylase, leading to tolerant traits [187]. MT–PA adjustment promotes the plant system’s antioxidant enzymatic apparatus, which regulates reactive species concentrations and ameliorates oxidative overexpressing damage reactions [188]. Moreover, MT-mediated PA catabolism has been demonstrated to be significant in limiting the over-generated reactive species formation under stressful events. The MT-induced regulation of PA metabolic flux promotes ionic stability in plant cells throughout stress environments [188]. According to research, MT–PA homeostasis involves interplay with SA, abscisic acid, and ethylene, resulting in a complex network of defense action under acute stress. It has been revealed that emphasis should be placed on the regulatory function of MT in the control of PA synthesis and the accompanying signaling components that provide plants with resistance to stress [35]. Novel hormonal crosstalks, including MT–PA modulation, have also been studied in hopes of acquiring a mechanistic understanding of the underlying processes. The study will enhance the relevant components of the usage of MT as well as PAs as polyfunctional phytoprotectants in the farming industry [189]. Thus, MT and PAs are effective for proper growth and yield under stressful environments. 

## 5. MT Interact with PGRs in Horticultural Plants

### 5.1. MT and AUXs 

Oxidative stress can occur due to various environmental factors by reducing total plant biomass, plant size, and photosynthetic pigments. MT and AUX, two important plant hormones, have been found to interact and work together in mitigating oxidative stress, a condition characterized by an imbalance between ROS production and the antioxidant defense system [123]. MT can regulate AUX biosynthesis and transport, influencing the distribution and accumulation of AUX in plant tissues. This MT-mediated regulation of AUX can impact the antioxidant defense system, enhancing the plant’s ability to cope with oxidative stress [123].

Many traits, i.e., the growth potential, rooting abilities, and gravitropism, were studied with both MT as well as AUX. The relationships between MT and indole acetic acid (IAA) molecules were investigated in lupine plants [145]. MT could also increase biomass production, as reported in earlier research [15]. MT levels and tissue type both influence growth. Root growth is more susceptible than leaf area. At elevated MT production, both IAA and MT inhibit development. Recent research has shown that high levels of MT inhibit IAA production while low levels of IAA boost it. The current research has shown that MT controls the function of signaling elements like AUX receptors, regulators, and small AUX-upregulated RNA genes to trigger growth [166]. Early lupine (*Lupinus albus* L.) work has also revealed that MT can promote root growth [145]. MT has now been shown to promote the formation of sideways and adventitious roots in a range of distinct species [15]. MT’s capacity to generate lateral and adventitious roots is among the most studied properties of this chemical, and it is commonly linked to AUXs [4]. MT efficiently enhanced the root system architecture and increased the mineral nutrient accumulation in tomato seedlings under cadmium toxicity [20]. Furthermore, MT positively regulated root growth and the development of tomato seedlings in tomato under cadmium toxicity [26]. MT is a potent molecule that modulates tomato root morphology. It significantly reduced embryonic root growth while encouraging the development and expansion of lateral roots, and it elevated the activation of numerous genes in the root tip cell and particular root components [190]. MT activities on the AUX transcription factor led to improvements in root physiology [190,191]. 

The figure highlights key physiological responses induced by MT supplementation, including enhanced antioxidant capacity, reduced oxidative damage, improved photosynthetic efficiency, and better osmotic regulation. These responses contribute to the improved stress tolerance and overall performance of horticultural crops under challenging environmental conditions.

### 5.2. MT and GA 

MT exogenously boosted the concentration of GA, which aided germination activities that were hampered by salinity in cucumber [192]. According to an earlier report on cucumber as well as red cabbage, MT possesses germination-promoting characteristics [192,193]. MT applied to *Brassica napus* L. subjected to salinity increased root growth by improving GA concentrations and increasing the expression of GA-generating enzymes. Furthermore, GA channels were increased, which express the soluble GA transmitter that binds with GA and DELLA proteins to form a complex that blocks DELLAs from inhibiting GA signaling. This causes high GA signaling, which promotes seedling development [166]. MT dosing increased GA levels and cyclin production in apple plants [194,195]. MT application along with GA significantly increased proline uptake, protected photosynthetic efficiency, upregulated antioxidant enzyme activity, and decreased oxidative damage in tomato seedlings under salinity toxicity [31]. The direct effect of MT on GA concentrations is still unclear. However, limited information revealed that MT increases GA concentration in reaction to different stressors [30]. The effects of MT on AUX and GA regulation for the modulation of the defense system focuses on sustainable crop production by mitigating oxidative injury under abiotic stressors (Table 5). 

### 5.3. MT and CKs 

Exogenous MT supplementation elevates CK concentrations. CK was found to stimulate the activation of MT heterologous proteins in response to a rise in endogenous MT [162]. Moreover, responder signaling pathways (ARR, types A and B) as well as other CK signaling genes were also activated. The combination of CK and MT increase physiological parameters such as photochemical effectiveness, total chlorophyll, and relative water content in drought-stressed wild species and isopentenyl transferase-overexpressing transgenic-spreading bent grass [196]. Similarly, in *B. napus*, overexpression of several CK signaling molecules, comprising A-ARR and B-ARR, was associated with increased growth after CK-mediated MT treatment [28]. CK and MT have also been related to sweet cherry ripening [197]. ABA and CK supplementation contributed to sustainable crop production (Table 6).

## 6. Conclusions and Future Concerns

MT acts as a signaling agent, regulating several physiological functions such as the photosynthetic system; balancing redox homeostasis; maintaining ion homeostasis; upregulating the AsA-GSH enzymes cycle; modulating secondary metabolites production, root development, seed germination; and regulating the response to environmental stresses. The ascorbates–glutathione cycle contributes to the tolerance mechanism of plants subjected to abiotic stressors (Figure 5). The combination of MT and other phytohormones in mitigating oxidative stress holds great potential for enhancing crop productivity and stress tolerance. MT acts as a powerful antioxidant, scavenging ROS and protecting cellular components from oxidative damage. Phytohormones such as AUXs, CK, and ethylene, in combination with MT, contribute to the regulation of antioxidant enzyme activities, gene expression, and physiological processes involved in stress responses. This interplay enhances the plant’s ability to counteract oxidative stress, improve growth, and increase crop yield. Harnessing the combined effects of MT and phytohormones provides promising avenues for sustainable horticultural crop production in the face of environmental challenges.

Conducting comprehensive studies to elucidate the specific mechanisms underlying the interplay between MT and different phytohormones, such as AUS, CK, and ABA, would provide valuable insights. Additionally, investigating the potential of exogenous application of MT and phytohormones as a combined treatment strategy could enhance stress tolerance and productivity in horticultural crops. Understanding these impacts could lead to the development of innovative approaches for sustainable crop management and improved resilience in the face of oxidative stress.

## Figures and Tables

**Figure 1 antioxidants-13-00663-f001:**
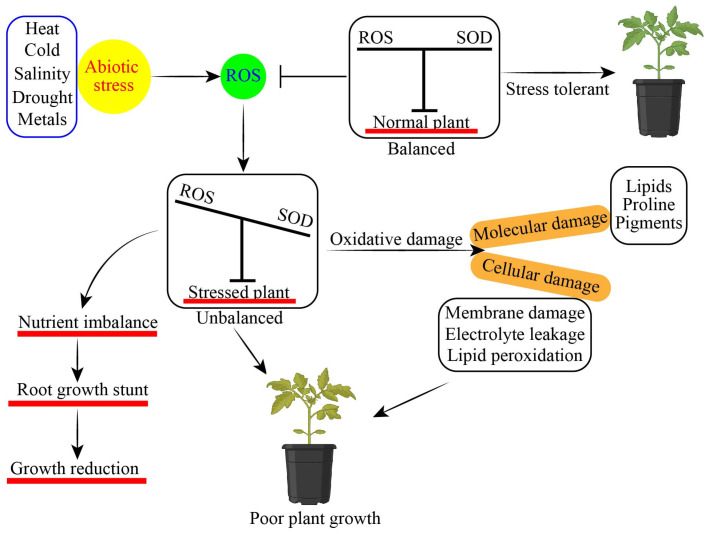
Effect of abiotic stress on plant growth and development of horticultural plants.

**Figure 2 antioxidants-13-00663-f002:**
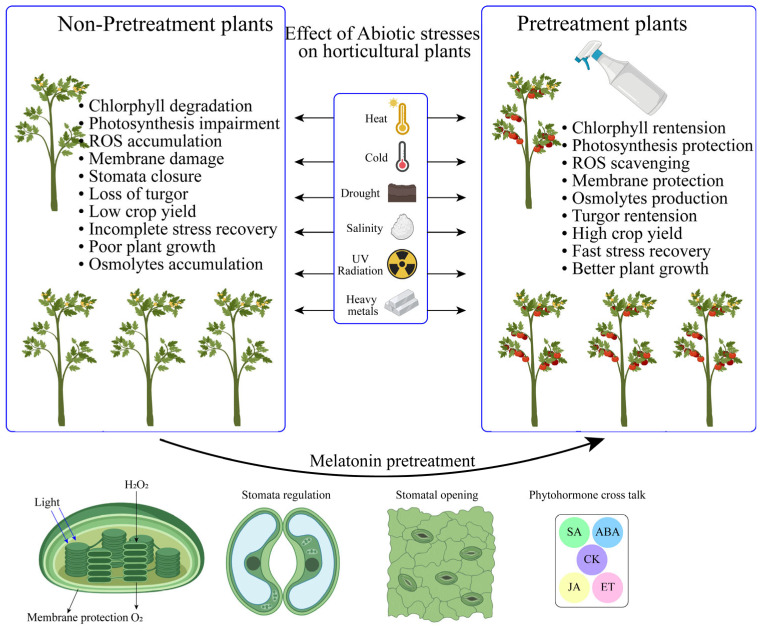
Mitigation of abiotic stress conditions via supplementation of melatonin on horticulture crops.

**Figure 3 antioxidants-13-00663-f003:**
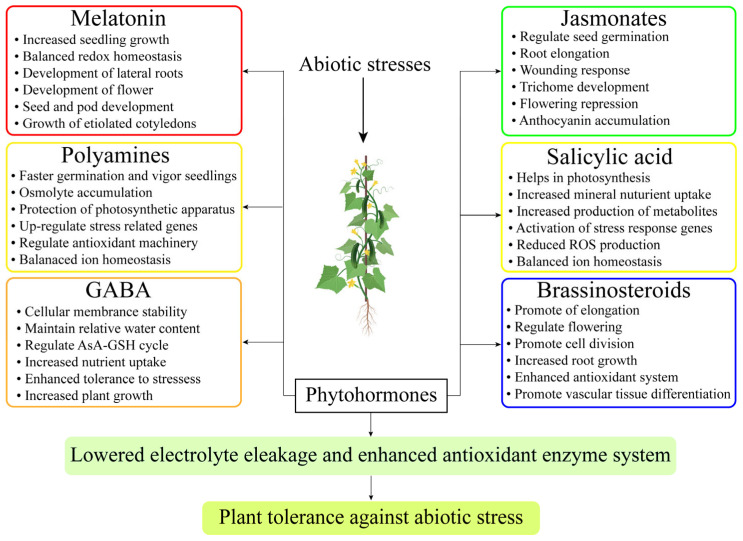
Supplementation of phytohormones for the mitigation of abiotic stressors in horticulture crops. The figure depicts the application of phytohormones, including melatonin, auxins, cytokinins, gibberellins, and abscisic acid, to crops under stress.

**Figure 4 antioxidants-13-00663-f004:**
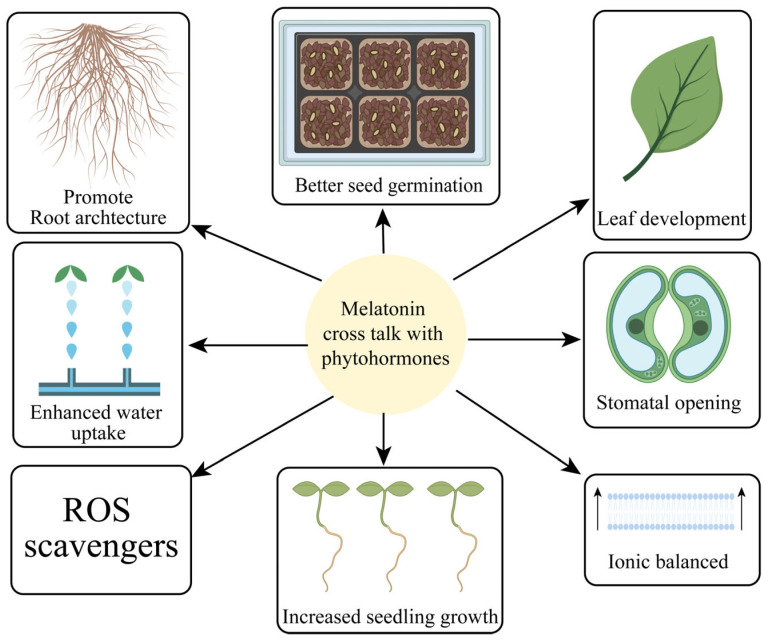
MT crosstalk with phytohormones for the regulation of photosynthesis and improved root architecture.

**Figure 5 antioxidants-13-00663-f005:**
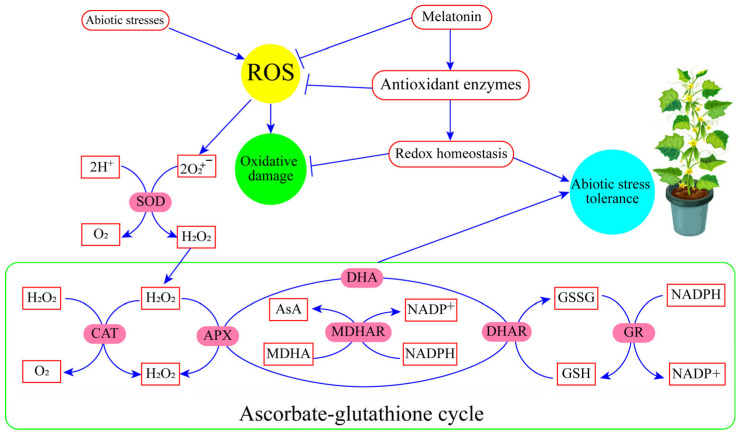
Contribution of the ascorbate–glutathione (AsA-GSH) cycle in abiotic stress regulation. The figure illustrates the key components and processes involved in the AsA-GSH cycle and its contribution to abiotic stress regulation in plants. The cycle consists of enzymatic reactions catalyzed by specific enzymes, including ascorbate peroxidase (APX), mono-dehydroascorbate reductase (MDHAR), dehydroascorbate reductase (DHAR), and glutathione reductase (GR).

**Table 1 antioxidants-13-00663-t001:** Adverse effects of abiotic stress on the sustainable production of horticultural crops.

Stress Type	Crops	Key Finding	Reference
Drought and salt	Apple	Stress tolerance improved via transcriptional factor *MdDREB2A*.	[45]
Drought	Tomato	About 70–75% of field capacity also increased the level of salt accumulation in the root zone of vegetables	[46]
Salinity	Cucumber	The application of 50 mM NaCl, which may be controlled by spraying various phytohormones on vegetable crops, had a negative effect.	[47]
Salinity	Lettuce	Lettuce performance was decreased because salinity resulted in the decrease of plant productivity.	[46]
HMs	Tomato	Overexpression of numerous genes like *StJAZ1* might reduce the relative leaf water capability in the vegetables.	[48]
Salinity	Apple	A salt-induced *MdCIPK6L* gene was isolated from an apple. Its expression was positively induced by abiotic stresses, stress-related hormones	[49]
Drought	Cucumber	The development, yield, and defense-related activities of the cucumber plants were expressly promoted by osmotic damage as well as oxidative stress.	[50]
Heat	Peppers	Temperature extremities are the major cause of proteins denaturing contributing in proper growth and yield.	[50]
Drought	Potato	Lipid peroxidation, hydrogen peroxide (H_2_O_2_), ROS, and malondialdehyde (MDA) were increased. Phytohormones can effectively lower lipid peroxidation and MDA generation.	[51]
Cold	Melon	The cooling stress conditions had a very negative impact on seed germination. Chiller injuries were lessened in part through CBF-responsive pathways.	[52]
Heat	Strawberry	Physiological, biochemical, and molecular basis were disturbed due to abiotic stressors.	[53]

**Table 2 antioxidants-13-00663-t002:** Potential function of MT in horticultural plants.

Functions	Reference
Regulated seed germination	[55]
Modulated Antioxidant defense mechanism	[22]
Maintained redox homeostasis	[56]
Reduced oxidative damage and membrane damage	[57]
Lowered electrolyte leakage and lipid peroxidation production	[36]
Enhanced photosynthetic machinery	[27]
Modified root system architecture	[13]
Boosted growth and development	[31]
Regulated lateral root development	[58]
Decreased HMs accumulation	[59]
Regulated fruit maturation	[56]
Upregulated secondary metabolites accumulation	[60]
Increased photosynthetic pigments content	[24]
Development of leaf size and plant height	[22]
Development of flowering	[61]
Vegetative growth promotion	[27]
Regulated production of phytohormones	[36]

**Table 4 antioxidants-13-00663-t004:** Effects of MT and ethylene focusing on the sustainable production of horticulture crops against abiotic stresses.

Crop Type	MT Spray	Major Effects	References
Tomato fruits	50 μM	It can induce stress-related responses, including stomatal closure, root growth inhibition, leaf senescence, and the activation of stress-responsive genes. MT treatment has been shown to downregulate the expression of ethylene biosynthesis genes, such as ACC synthase and ACC oxidase, leading to reduced ethylene production. Ethylene receptor genes were expressed.	[126]
Tomato plants	0.1 mM	Increase in the potential of ascorbic acid and lycopene contents. Elevated quality and yield of tomato fruits.	[158]
Cassava roots	100 μM	Downregulation of ethylene biosynthesis genes by MT may contribute to the mitigation of stress-induced ethylene responses in plants. Reduction in the post-harvest losses. Increase in the shelf life of tuberous. Improved plant defense system against biotic and abiotic stress.	[125]
Peach fruits	0.1 mM	Reduction in post-harvest senescence. Increase in fruit firmness and flavor. Reduction in physiological weight losses.	[159]
Strawberry fruits	100 μM	Reduction in the post-harvest senescence. Increase in the ATP, antioxidants, and shelf life. Improved potential of defense system against pathogens.	[53]

**Table 5 antioxidants-13-00663-t005:** Effects of MT on AUXs and GA, focusing on sustainable crop production under abiotic stress.

Crop Type	MT Spray	Major Effects	References
Tomato	12.5–100 μM	IAA increased up to two-fold. AUX carrier proteins were significantly activated. Increase of IAA signaling molecules. Drought tolerance improved in the plants.	[126]
Tomato overexpressing of serotonin *N-acetyltransferase*	0.2 μM	Abiotic stress tolerance was proved by modifications in plant physiological mechanisms. Improvement in the IAA up to 7-fold.	[126]
Banana	0.01–0.5 μM	IAA up to 2-fold was found to be induced. Root morphology was improved.	[40]
Pepper seedling	5 μM	MT enhanced seedling growth; Reduced oxidative damage and upregulated the antioxidant enzyme system	[20]
Pepper seedlings	5 μM	MT protected leaf photosynthetic efficiency, enhanced mineral nutrient accumulation, and decreased heavy metal uptake	[20]
Cucumber seedlings	1 μM	GA biosynthesis was improved. Tolerance against salinity was enhanced by modulation of physiological and biochemical processes. An increase in germination rate was recorded.	[123]

**Table 6 antioxidants-13-00663-t006:** Effects of MT interplay with ABA and CK on the sustainable production of horticulture crops.

Crop Type	MT Spray	Major Effects	References
Watermelon plants	1.5–150 μM	MT and CK can coordinately modulate the expression of stress-related genes, including those involved in antioxidant defense, osmotic regulation, and stress signaling pathways. The activation of cold-responsive genes was recorded in plants. Plant tolerance was improved against cold. Activation of IAA, GAs, ethylene, and JA-related genes. ABA receptor PYL8 gene expression.	[198]
Apple leaves	10–500 μM	ABA endogenous level was restricted. Reduced ABA biosynthesis genes. Increase in ABA catabolism genes.	[148]
Cucumber seedlings	1 μM	The ABA endogenous level was suppressed. Reduced ABA biosynthesis occurs. Increase in the ABA catabolism.	[192]
Ryegrass	20 μM	Photosynthesis potential was improved. Cell membrane stability was regulated. CKs endogenous level enhanced. Expression of CKs biosynthesis genes. Activation of CKs signaling transcription factors.	[33]

## Data Availability

No new data were created or analyzed in this study. Data sharing is not applicable to this article.

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
