# Peer review of "Melatonin Interaction with Other Phytohormones in the Regulation of Abiotic Stresses in Horticultural Plants"

_antioxidants, 2024, doi:10.3390/antiox13060663_

Round 1
Reviewer 1 Report
The Manuscript submitted by Huang and Jin, Manuscript titled “Melatonin and plant hormone Synergy: Key Influences on im-2 provement in Horticulture Crops” The author tried to compiled the compressive information on the role of melatonin and interaction with other phytohormone” Author collected a significant and novel information. I think manuscript is suitable for this well reputed journal, but made final decision, I have some minor correction.Comment to author Line 7: a significant threat Line 14: biomolecule Line 17-18: revise Revise keywords, Fig.? or figure? Use according to format Symmetry Line 104, first time define full name “BRs, MEL, SA, JA” Line 108-109: revise this line Line 116: BRS, should be small Check Table 1: row 2, remove typo “Up to 6-fold “used the concentration in micromole Line 182: use full name in the legend “MT” Line 243: NO? First time define Line 260: JAs can be used to control environmental risks, remove this line Line 283: only first time define, not define repeatedly “Jasmonates (JAs)” Line 340: check this combine term, “phytohormones Polyamines (PAs) group” Line 373: PA catabolism Line 384: MT and PAs Table 4, row first” ABA stress? Check this term Line 60: “horticultural crop production during environmental challenges. Important: Author check abbreviation list and define at first use.
Author Response
Major comments
The Manuscript submitted by Huang and Jin, Manuscript titled “Melatonin and plant hormone Synergy: Key Influences on im-2 provement in Horticulture Crops” The author tried to compiled the compressive information on the role of melatonin and interaction with other phytohormone” Author collected a significant and novel information. I think manuscript is suitable for this well reputed journal, but made final decision, I have some minor correction.
Ans: Thanks for the comments. Correction has been made and highlighted in red.
Detail comments
Comment to author Line 7: a significant threat Line 14: biomolecule Line 17-18: revise
Ans: All the corrections are highlighted in red and easy to find at the suggested lines 7,14,17-18.
Revise keywords, Fig.? or figure? Use according to format Symmetry Line 104, first time define full name “BRs, MEL, SA, JA” Line 108-109: revise this line Line 116:
Ans: Keywords are revised. Line 104 corrected and highlighted in red. All the abbreviations are well elaborated at their suitable place.
BRS, should be small Check Table 1: row 2, remove typo “Up to 6-fold “used the concentration in micromole Line 182: use full name in the legend “MT” Line 243: NO? First time define Line 260:
Ans: Table 1 has been corrected. Typo mistakes have been resolved.
JAs can be used to control environmental risks, remove this line Line 283: only first time define, not define repeatedly “Jasmonates (JAs)” Line 340:
Ans: Repetition has been corrected.
check this combine term, “phytohormones Polyamines (PAs) group” Line 373: PA catabolism Line 384:
Ans: Phytohormones Polyamines (PAs) group corrected as phytohormones such as polyamines (PAs) group.
MT and PAs Table 4, row first” ABA stress? Check this term Line 60: “horticultural crop production during environmental challenges. Important: Author check abbreviation list and define at first use.
Ans: All the corrections are highlighted in red and easy to find at the suggested lines.
Reviewer 2 Report
The review proposed by Shanxia Huang and Songheng Jin presents an overview of phytomelatonin and the already established plant hormones, including the cross-talk between them and involvement in mitigating abiotic stress in horticulture crops. The good aspect is that the subject is an important and intriguing one, that still needs a lot of research to improve our understanding and optimize the treatments.
Unfortunately, there are some major drawbacks as well.
The first one is that the authors do not present the difference between their review and the other recent ones on this topic. I give just some examples:
https://www.sciencedirect.com/science/article/pii/S2213231723002069
https://www.mdpi.com/2073-4425/13/10/1699
https://www.frontiersin.org/journals/plant-science/articles/10.3389/fpls.2022.1095363/full
https://www.tandfonline.com/doi/full/10.1080/15592324.2015.1096469
The authors should rewrite their paper highlighting the novelty of this review and why it is necessary to have an additional review.
The second concern is about the organization of the paper:
- why presenting the biosynthesis of melatonin at the end of the paper, in chapter 6? And why mentioning about microorganisms suddenly, when there is no mention about this until that point. The more logical order is to start with the biosynthesis and remove the subject of microorganisms
- many parts of the paper either repeat the same Information in a different form or jump from one idea to another without any apparent connection, or simply mention something else that was not related to the neighboring text. Some examples, as it is difficult to present here all:
lines 82-95 (too many sentences mentioning abiotic stress and the effects on plants - practically saying more or less the same thing; please be more concise)
lines 118-122 (after just a few general lines about MT and established plant hormones and one sentence about abiotic stress, the authors conclude with "so" that "it has been studied that...is complicated" (the phrase is not logical) and that they present detailed insights in in Table 1 (which is far from being detailed or having many references)
lines 229-231 (after presenting the positive effects of supplementation with SA, the authors write "Furthermore..." and present the the reduction of biomass etc because of stress. It does not make sense.
lines 258-263 (many sentences that either repeat the same idea or jump to a general remark that it is not necessarily related to the specific subject (JAs) of the other sentences ("Hence, plants have evolved their defense mechanisms in response to environmental challenges such as waterlogging in peppers" - it can fit anywhere.)
Lines 270-271 (seed priming in general? What is the connection with the previous text?)
Lines 283-286 ("Jasmonates (JAs) are plant hormones that include methyl jasmonate (MeJA) and jasmonic acid (JA). They regulate a wide spectrum of crop productivity and stress responses. Hence JAs increase the expression of genes, proteins, and bioactive molecules involved in the plant defense system" - this should be mentioned in the beginning of the section. Moreover, why use the term "hence", as the sentence does not come as a conclusion)
lines 334-336 (it is not properly discussed the relation between skotomorphogenesis and brassinosteroids).
lines 448-458 (many various Information about H2O2, stomatal closure, MAPK, multifunctional non-toxic amphiphlic agent (this phrase is not clear), receptors, CAND2, antioxidant in just several phrases, without proper explanation of the terms, and the links between all these)
- chapter 5: what is the relevance to the subject of the paper? There is no clearly presented connection.
All in all, I find very difficult to follow this review and, based on my comments above, I suggest a proper rethinking and rewriting of the paper, arguing for the novelty and presenting the Information in a coherent manner.
There are many aspects that should be improved. I mentioned some at the Major comments.
Other observations:
The abbreviations are used without giving the full name first time they appear and then using the abbreviations
Lines 139-140: what is the different method and what do you mean by future investigations?
Line 186: what do you mean "producing genes". I do not think the term is correct, the genes are not produced. Maybe you want to say "expressing genes" (and these should be mentioned).
Line 197: not clear why the direct effect of MT on GA is disputed because it was shown to increase GA concentrations.
Line 204: MT heterologous proteins? Please write properly.
Line 206: Cytokinin and MT behave together? Probably you want to say "act"
Lines 270-271: reference should be given
Line 289: JA proteins (please do not use this kind of expression, but rather give the full explanation)
Lines 290-291: "So, the plant's response to JA is reduced which improves abiotic tolerance" - please reconsider this afirmation or develop, because if confusing
Line 293: "Chromium toxicity poorly decreased the growth of tomato seedlings" - what do you mean by "poorly"?
Lines 317-318 (The overproduction of ROS, MDA, H2O2 - are you sure?)
Lines 322-323 ("BRs are considerably more successful at coping with abiotic stress because cucumbers have been found to have a comparable tolerance process" - the relation between cucumbers and BRs is not explained)
Lines 325-327 ("It is considerably more effective for crops with better yields that are growing in both normal and abiotic stress situations." - not clear what is more effective and why for crops with better yields)
Line 352: what do you mean about the polarization of ions? Maybe you want to refer to the polarization of membranes?
Line 362: what do you mean by electron movement? Do you mean "electron transport chain"?
Line 368: is it "crops" instead of "cops"?
Lines 399, 401: references should be provided
Lines 421-423: lowering putrescine levels is in contradiction with "improve the number of polyamines produced" (number is not the right word)
Line 433: Not clear what you mean by "absence of knowledge", as recently there are several studies about MT receptors and you even mention something about MT1 and MT2. Therefore this part should be clear. What lack of knowledge makes the specific function and signaling mechanism unclear and what should be investigated in the future?
Lines 454-458 - the authors should first discuss properly about the receptor CAND2 and provide references. Next, this part should be developed and the last sentence should be clearly explained, as the concern about the overall function of MT as potent antioxidant is not evident.
Table 2 - there is no Information about supplementation with abscisic acid and cytokinin (see lines 212-213), only MT spray.
Table 3 - the title and line 244 mention about the effects of ethylene, but the table presents only the effects on ethylene, which is not the same thing and it generates confusion.
Author Response
Does the title describe the article's topic with sufficient precision, bearing in mind that it is a review article?
The title reflects most of the content of the review, but there are two aspects that are not clear: 1. what do authors wanted to mean by using the term "synergy". The authors should be careful in using this term, as the definition of synergy is: "the interaction or cooperation of two or more organizations, substances, or other agents to produce a combined effect greater than the sum of their separate effects." From the Information provided in the review, this synergism is not really relevant. Moreover, quantitative data on doses should be provided to demonstrate synergy, which has not been yet performed to the best of my knowledge. Even less we can talk about synergism in the case of endogenous levels, where the situation is much more complicated. I suggest the authors change "synergy" with "cross-talk". 2. It is not clear how the title covers the content of section 5 "Molecular implications in the development of tolerant germplasm", which has 120 lines of which just the last 7 and Table 5 refer to melatonin and (other) phytohormones. It is anyway not clear what was the purpose of section 5. Moreover, the authors should decide if they have arguments (which should be presented and discussed) to consider melatonin as a phytohormone, in which case the title should be : "and other plant hormones", or, if not, either to remove melatonin from Table 5 or better change the title of Table 5 to include all molecules (ascorbic acid, polyamines as well).
Ans: Title has been revised.
Does the abstract/introduction provide a sufficiently clear description of the topic subject of this review?
Same as for the title, section 5 is not reflected in the abstract and very little in the introduction, and the connection between germplasm, genetic techniques, and phytohormones is not properly presented, in neither abstract, introduction, nor section 5. Moreover the abstract presents mainly the effects of melatonin, but each chapter describes the effects of the other molecules independenlty, and this is not reflected in the abstract.
Are the conclusion supported by results?
The authors mention several times about synergy. As commented for the title, unless there is a clear demonstration for the claimed synergism, this term should be avoided.
Ans: As per reviewer corrections, word synergy replaced with more scientific word “combined”. Hope it will be fine now.
Are the timeliness, breadth and accuracy of the discussion qualified?
The review is presented and discussed in a rather chaotic manner, and it is difficult to follow. There are many jumps from one idea to another without explaining the connection. I did not understand the purpose of section 5 and its fit within the review. I will detail in the general comments.
Ans: Author agreed with reviewer comments. However, development of tolerant germplasm is also interesting by using molecular implications.
Are all cited of the references relevant to the research?
I mention again that the scope of section 5 is not clear, and in consequence the references cited in there.
Ans: Citations of the section have been imporved as per reviewer suggestions
Does this article provide a relevant contribution to the scientific discussion of this topic?
The novelty / difference of this review in comparison with other similar reviews that have been published is not clear. Extensive editing of English language required
Ans: Author tried their best to improve the work novelty. However, English corrections are made by subject expert.
Major comments
The review proposed by Shanxia Huang and Songheng Jin presents an overview of phytomelatonin and the already established plant hormones, including the cross-talk between them and involvement in mitigating abiotic stress in horticulture crops. The good aspect is that the subject is an important and intriguing one, that still needs a lot of research to improve our understanding and optimize the treatments.
Unfortunately, there are some major drawbacks as well.
The first one is that the authors do not present the difference between their review and the other recent ones on this topic. I give just some examples:
https://www.sciencedirect.com/science/article/pii/S2213231723002069
https://www.mdpi.com/2073-4425/13/10/1699
https://www.frontiersin.org/journals/plant-science/articles/10.3389/fpls.2022.1095363/full
https://www.tandfonline.com/doi/full/10.1080/15592324.2015.1096469
The authors should rewrite their paper highlighting the novelty of this review and why it is necessary to have an additional review.
Ans: most of the previous literature focus on melatonin. However, limited work is available is one melatonin and phytohormones.
The second concern is about the organization of the paper:
- why presenting the biosynthesis of melatonin at the end of the paper, in chapter 6? And why mentioning about microorganisms suddenly, when there is no mention about this until that point. The more logical order is to start with the biosynthesis and remove the subject of microorganisms
Ans: Section 6 has been corrected as “Melatonin biosynthesis as a tryptophan derivative in plants” by deleting microorganisms and its related information.
- many parts of the paper either repeat the same Information in a different form or jump from one idea to another without any apparent connection, or simply mention something else that was not related to the neighboring text. Some examples, as it is difficult to present here all: lines 82-95 (too many sentences mentioning abiotic stress and the effects on plants - practically saying more or less the same thing; please be more concise).
Ans: Repeated information has been deleted from line 82-95.
lines 118-122 (after just a few general lines about MT and established plant hormones and one sentence about abiotic stress, the authors conclude with "so" that "it has been studied that...is complicated" (the phrase is not logical) and that they present detailed insights in in Table 1 (which is far from being detailed or having many references)
Ans: Summarized information at line 118-122 was just to highlight the table.
lines 229-231 (after presenting the positive effects of supplementation with SA, the authors write "Furthermore..." and present the reduction of biomass etc because of stress. It does not make sense.
Ans: Correction has been made and lines moved at line 227.
lines 258-263 (many sentences that either repeat the same idea or jump to a general remark that it is not necessarily related to the specific subject (JAs) of the other sentences ("Hence, plants have evolved their defense mechanisms in response to environmental challenges such as waterlogging in peppers" - it can fit anywhere.)
Ans: General information of line 258-263 has been deleted.
Lines 270-271 (seed priming in general? What is the connection with the previous text?)
Ans: Seed priming information is omitted to maintain the continuity of the review article.
Lines 283-286 ("Jasmonates (JAs) are plant hormones that include methyl jasmonate (MeJA) and jasmonic acid (JA). They regulate a wide spectrum of crop productivity and stress responses. Hence JAs increase the expression of genes, proteins, and bioactive molecules involved in the plant defense system" - this should be mentioned in the beginning of the section. Moreover, why use the term "hence", as the sentence does not come as a conclusion).
Ans: Lines 283-286 have been moved as per reviewer suggestions. Term “hence” has been deleted.
lines 334-336 (it is not properly discussed the relation between skotomorphogenesis and brassinosteroids).
Ans: Correction has been made at line 328-330.
lines 448-458 (many various Information about H2O2, stomatal closure, MAPK, multifunctional non-toxic amphiphlic agent (this phrase is not clear), receptors, CAND2, antioxidant in just several phrases, without proper explanation of the terms, and the links between all these).
Ans: some corrections have been made and highlighted in red.
- chapter 5: what is the relevance to the subject of the paper? There is no clearly presented connection. All in all, I find very difficult to follow this review and, based on my comments above, I suggest a proper rethinking and rewriting of the paper, arguing for the novelty and presenting the Information in a coherent manner.
Ans: Thanks for the correction. Molecular implications are important for development and identification of stress tolerant germplasm.
Detail comments
There are many aspects that should be improved. I mentioned some at the Major comments.
Other observations:
The abbreviations are used without giving the full name first time they appear and then using the abbreviations
Lines 139-140: what is the different method and what do you mean by future investigations?
Ans: At line 137-138. Sentence has been rephrased.
Line 186: what do you mean "producing genes". I do not think the term is correct, the genes are not produced. Maybe you want to say "expressing genes" (and these should be mentioned).
Ans: Highlighted section is revised and correct at line 183-184.
Line 197: not clear why the direct effect of MT on GA is disputed because it was shown to increase GA concentrations.
Ans: Correction has been made as per suggestions.
Line 204: MT heterologous proteins? Please write properly, Line 206: Cytokinin and MT behave together? Probably you want to say "act", Lines 270-271: reference should be given
Ans: Highlighted information has been incorporated and highlighted in red.
Line 289: JA proteins (please do not use this kind of expression, but rather give the full explanation), Lines 290-291: "So, the plant's response to JA is reduced which improves abiotic tolerance" - please reconsider this afirmation or develop, because if confusing.
Ans: Correction has been made as per suggestions.
Line 293: "Chromium toxicity poorly decreased the growth of tomato seedlings" - what do you mean by "poorly"?
Ans: Poorly replaced with reduced.
Lines 317-318 (The overproduction of ROS, MDA, H2O2 - are you sure?)
Ans: Correction has been made as per suggestions.
Lines 322-323 ("BRs are considerably more successful at coping with abiotic stress because cucumbers have been found to have a comparable tolerance process" - the relation between cucumbers and BRs is not explained).
Ans: Line 314-316. Correction has been made as per suggestions.
Lines 325-327 ("It is considerably more effective for crops with better yields that are growing in both normal and abiotic stress situations." - not clear what is more effective and why for crops with better yields).
Ans: Correction has been made as per suggestions.
Line 352: what do you mean about the polarization of ions? Maybe you want to refer to the polarization of membranes?
Ans: At line 344-345. Correction has been made as per suggestions.
Line 362: what do you mean by electron movement? Do you mean "electron transport chain"?
Ans: Line 353-354. electron movement corrected as membrane leakage.
Line 368: is it "crops" instead of "cops"?
Ans: cops replaced with crops
Lines 399, 401: references should be provided
Ans: Reference has been incorporated.
Lines 421-423: lowering putrescine levels is in contradiction with "improve the number of polyamines produced" (number is not the right word).
Ans: Number of polyamines produced corrected as polyamines production
Line 433: Not clear what you mean by "absence of knowledge", as recently there are several studies about MT receptors and you even mention something about MT1 and MT2. Therefore this part should be clear. What lack of knowledge makes the specific function and signaling mechanism unclear and what should be investigated in the future?
Ans: Correction has been made as per suggestions.
Lines 454-458 - the authors should first discuss properly about the receptor CAND2 and provide references. Next, this part should be developed and the last sentence should be clearly explained, as the concern about the overall function of MT as potent antioxidant is not evident.
Ans: Correction has been made as per suggestions.
Table 2 - there is no Information about supplementation with abscisic acid and cytokinin (see lines 212-213), only MT spray. Table 3 - the title and line 244 mention about the effects of ethylene, but the table presents only the effects on ethylene, which is not the same thing and it generates confusion.
Ans: Available information is included in the tables.
Round 2
Reviewer 2 Report
Although the authors made some improvements to the manuscript, the main concerns have not been addressed properly. For example, the novelty of the review compared with the previous reviews is still not clearly presented. Section 5 is not properly presented, as I have commented above. Section 6 was reduced significantly but I do not understand why to present this short section at the end of the review. To me it makes more sense to present it before chapter 3.
Another problem is that I cannot easily follow the changes, because the authors did not use track changes. Please use track changes so that I can see the differences clearly.
Lines 118-121 still need improvement: "it was shown" or "it was proven" or "it was revealed" that "the interplay....is complicated". Please delete "So"
The lines 229-231, were not improved as expected. If you talk in the previous sentences about the positive effects of SA, you cannot jump to the negative effects of stress (not to mention that you even add "Furthermore"). Please present in the logical order. First you must talk about the effect of stress and next about how SA etc. mitigated the stress, if the case.
lines 448-458, I cannot see the changes highlighted in red. Please indicate the new lines.
lines 453-471, the provided iThenticate report indicates that almost the entire paragraph is highly similar (almost identical) to the text from this article: https://link.springer.com/chapter/10.1007/978-981-99-6741-4_3. Please rephrase this part to reduce the similarity.
I mentioned that "the abbreviations are used without giving the full name first time they appear and then using the abbreviations", but there was no answer.
Please use track changes for all the modifications made (also the previous ones) so that I can easily follow them.
Author Response
Major comments
Comment: Although the authors made some improvements to the manuscript, the main concerns have not been addressed properly. For example, the novelty of the review compared with the previous reviews is still not clearly presented. Section 5 is not properly presented, as I have commented above. Section 6 was reduced significantly but I do not understand why to present this short section at the end of the review. To me it makes more sense to present it before chapter 3.
Answer: Thank you for your valuable feedback. We acknowledge the importance of clearly highlighting the novelty of our review compared to previous ones. In this revision, we have made further efforts to address this concern and ensure that the manuscript content aligns more closely with the title and abstract.
Regarding your comment about sections 5 and 6, we have taken your feedback into consideration and have remove both sections. We agree that they did not appropriately complement the overall structure of the manuscript. Additionally, we have significantly expanded upon the existing content and incorporated new information to better align with the title and abstract.
We appreciate your patience and understanding as we work to improve the manuscript, and we are committed to addressing all concerns to enhance the quality of our work.
Comment: Another problem is that I cannot easily follow the changes, because the authors did not use track changes. Please use track changes so that I can see the differences clearly.
Answer: Thank you for your feedback. We understand your concern regarding the use of track changes and apologize for any inconvenience caused by not implementing it. We acknowledge that it can aid in understanding the revisions more clearly. In the revised version, we have utilized red color to highlight the sections that have been updated. Given that a substantial amount of new material has been incorporated while deleting previous content, this approach was chosen over word-by-word tracking to prevent confusion and ensure clarity.
Detail comments
Lines 118-121 still need improvement: "it was shown" or "it was proven" or "it was revealed" that "the interplay....is complicated". Please delete "So"
Response: We have revised this sentence and also corrected the whole section.
The lines 229-231, were not improved as expected. If you talk in the previous sentences about the positive effects of SA, you cannot jump to the negative effects of stress (not to mention that you even add "Furthermore"). Please present in the logical order. First you must talk about the effect of stress and next about how SA etc. mitigated the stress, if the case.
Response: We have thoroughly revised the sentence in question, addressing all typos and restructuring it to align with your suggestion. Additionally, we have reworked this section to ensure logical coherence between discussing the effects of stress and then detailing how SA and related factors mitigate these effects. Thank you for bringing this to our attention, and we appreciate your guidance in refining the manuscript.
lines 448-458, I cannot see the changes highlighted in red. Please indicate the new lines.
Response: Section 5 has been removed, and we have accurately revised all other sections of the manuscript.
lines 453-471, the provided iThenticate report indicates that almost the entire paragraph is highly similar (almost identical) to the text from this article: https://link.springer.com/chapter/10.1007/978-981-99-6741-4_3. Please rephrase this part to reduce the similarity.
Response: We have removed this part and added new relevant material.
I mentioned that "the abbreviations are used without giving the full name first time they appear and then using the abbreviations", but there was no answer.
Response back: We now addressed your concern by revising the abbreviation list. Now, each abbreviation is defined the first time it is used in the manuscript.
Please use track changes for all the modifications made (also the previous ones) so that I can easily follow them.
Response: We understand your concern regarding the use of track changes and apologize for any inconvenience caused by not implementing it. We acknowledge that it can aid in understanding the revisions more clearly. In the revised version, we have utilized red color to highlight the sections that have been updated. Given that a substantial amount of new material has been incorporated while deleting previous content, this approach was chosen over word-by-word tracking to prevent confusion and ensure clarity.
Does the title describe the article's topic with sufficient precision, bearing in mind that it is a review article?
I am sorry, but my comment on how does the title cover section 5 has not been answered. Or to be more precise, chapter 5 is not presented properly and therefore it seems to be outside of the subject suggested by the title. I understand that the molecular implications are important, but most of chapter 5 does not make the correlation between the molecular aspects presented and MT and the other phytohormones (the subject of the paper). You just discuss molecular aspects in general and you do not present the relation between these molecular aspects, the techniques and the phytohormones. What does CRISPR/Cas, QTL etc. have to do with the phytohormones? You present this Information (2 pages and a half) completely independent of phytohormones. Why presenting this Information? Please highlight in chapter 5 the relevance of the presented molecular aspects to phytohormones. Otherwise the title does not describe the article content with sufficient precision.
Response: We appreciate your detailed feedback regarding the alignment of Chapter 5 with the overall theme suggested by the title. Recognizing your concerns, we have made the decision to remove both sections 5 and 6 from the draft. We understand that these sections did not integrate well with the remaining content and may have deviated from the primary focus of the manuscript. Thank you for bringing this to our attention, and we have revised the entire draft to ensure that it aligns more closely with the title and abstract, as per your suggestion. We are committed to improving the clarity and coherence of our work and appreciate your guidance in this process.
Does the abstract/introduction provide a sufficiently clear description of the topic subject of this review?
The authors did not answer my comments
Response: We sincerely apologize for any inconvenience caused. We have thoroughly revised the manuscript title, abstract, and introduction section to address your concerns comprehensively. We hope that these revisions have significantly improved the overall clarity and alignment with the intended content.
Are the timeliness, breadth and accuracy of the discussion qualified?
Comments: I am sorry, but is your paper about phytohormones or about anything that is interesting? If tolerant germplasm, its molecular aspects and the techniques are interesting, please connect them properly to the main subject of the paper, phytohormones. In the abstract, in introduction and chapter 5. Otherwise change the title to include these aspects as well.
Response: Thank you for your feedback. We have revised the title to encompass all relevant aspects of the paper. Additionally, we have removed Chapter 5 from the manuscript. We appreciate your input and are committed to ensuring that the paper aligns cohesively with its title and subject matter.
The comment was not properly answered. It is anyway difficult to follow as the authors did not use track changes. Please improve chapter 5 as mentioned above in order to justify the references.
Response: We have taken steps to ensure that all references cited in the manuscript are relevant to the research. Additionally, we have removed any unnecessary references and added suitable ones according to the content of the manuscript.
I cannot follow how and where the authors indicated the novelty/difference of this review compared with other similar reviews. Please write in your manuscript about previous similar reviews (about phytohormones), what is different in this review and why your review is important for the scientific community. Please mention the lines where I can find the changes, not just that you tried to improve.
Response: Thank you for your insightful comment. We have taken your feedback into consideration and have made several revisions to address it. Specifically, we have added lines highlighting the novelty of our review compared to other similar reviews at the end of the introduction section. Additionally, we have included discussions on recent research regarding melatonin and its interaction with other phytohormones.
Round 3
Reviewer 2 Report
The authors changed almost half of the review and now seems to be more coherent.
But, as mentioned at Recommendations, the specification of the novelty of the review compared to the previous reviews on this subject is still missing. So, it is hard to find the importance of the contribution to the scientific community.
It is in the authors' interest to highlight the differences between this review and other reviews on this topic. When I say this topic, I mean melatonin and other phytohormones, not related subjects.
Please do not try to avoid these other reviews. You need to cite them in your paper.
As mentioned twice already until now, the Introduction should provide the justification for another review on melatonin and other hormones, by mentioning the differences. Every review made, as well as any other type of article, no matter on which subject should bring something new.
The authors claim they did this, but I cannot see. If you did, please indicate in your comments the corresponding lines where I can find this Information.
I am sorry, but I cannot agree with acceptance for publication until I see this comparison clearly.
I do not have any other detail comments.
Author Response
The authors changed almost half of the review and now seems to be more coherent.
But, as mentioned at Recommendations, the specification of the novelty of the review compared to the previous reviews on this subject is still missing. So, it is hard to find the importance of the contribution to the scientific community.
It is in the authors' interest to highlight the differences between this review and other reviews on this topic. When I say this topic, I mean melatonin and other phytohormones, not related subjects.
Response back: We have added substantial information to the introduction to highlight the significant differences between our review and previous reviews on the same topic. This emphasizes the novelty of our work in comparison to other reviews focused on melatonin and other phytohormones.
Please do not try to avoid these other reviews. You need to cite them in your paper.
Response back: As per our knowledge, recent special issues on melatonin have been published in journals such as Scientia Horticulturae, South African Journal of Botany, Frontiers in Plant Science, and IJMS. We have reviewed and cited articles from these special issues, with publications dating to the end of 2023 and the start of 2024. Specifically, we have cited around 10 articles from early 2024 and over 20 articles from late 2023. This demonstrates our effort to incorporate the most recent literature in our manuscript.
As mentioned twice already until now, the Introduction should provide the justification for another review on melatonin and other hormones, by mentioning the differences. Every review made, as well as any other type of article, no matter on which subject should bring something new. The authors claim they did this, but I cannot see. If you did, please indicate in your comments the corresponding lines where I can find this Information.
Response: We have added substantial information and made efforts to comply with the significant details. Please refer to the end section of the introduction for these updates.
I am sorry, but I cannot agree with acceptance for publication until I see this comparison clearly.
Response: We appreciate your feedback and encouragement, such as your recent comment that "the authors changed almost half of the review, and now it seems to be more coherent." We have made substantial revisions to our manuscript following your guidance. We hope this addresses your concerns, and we thank you for your understanding.